# Thermodynamics of organic electrochemical transistors

**Matteo Cucchi** [1,2] ✉, **Anton Weissbach** [2]**, Lukas M. Bongartz**[2],
**Richard Kantelberg** [2]**, Hsin Tseng** [2]**, Hans Kleemann** [2] **& Karl Leo**[2]

Despite their increasing usefulness in a wide variety of applications, organic electrochemical transistors still lack a comprehensive and unifying physical framework able to describe the current-voltage characteristics and the polymer/electrolyte interactions simultaneously. Building upon thermodynamic axioms, we present a quantitative analysis of the operation of organic electrochemical transistors. We reveal that the entropy of mixing is the main driving force behind the redox mechanism that rules the transfer properties of such devices in electrolytic environments. In the light of these findings, we show that traditional models used for organic electrochemical transistors, based on the theory of field-effect transistors, fall short as they treat the active material as a simple capacitor while ignoring the material properties and energetic interactions. Finally, by analyzing a large spectrum of solvents and device regimes, we quantify the entropic and enthalpic contributions and put forward an approach for targeted material design and device applications.

Organic mixed ion-electron conductors (OMIECs)[1,2] emerged as an ideal class of materials for bioelectronic applications, both in-vivo and in-vitro. The attention to OMIECs stems from a reversible and energy-efficient redox mechanism that allows controlling the charge carrier density of the material. These appealing features, absent in traditional organic and inorganic electronic devices, have been harnessed to design human-compatible electronics, e.g., circuits that can be implanted within an organism and interact with tissues, neurons, blood, etc.[3–5]. The most prominent device based on OMIECs is the organic electrochemical transistor (OECT)[6], whereby a gate electrode is employed to flood or deplete the OMIEC with ionic species, therefore switching on and off its conductivity. The traditional OECT architecture, sketched in Fig. 1a, has been employed for a broad spectrum of applications of technological interest as proven by the number of successful demonstrations of OECT-based devices for biosensors[7], artificial synapses[8,9], hardware capable of computing[10–12], and wearable electronics for biofluid and biosignal monitoring[3,13,14]. More recently, the application scope of OECTs has been further expanded, entering *dry* electronics: using solid electrolytes, OECT-based devices were shown to be an excellent candidate for non-volatile memories[15] as well as a good alternative to traditional low-cost organic

and inorganic digital circuitry[16,17]. This multifaceted range of applications demands a detailed understanding of the physical and chemical processes underlying the OECT operation, as well as comprehensive electronic modeling. In their seminal paper, Bernards et al. borrowed the mathematical framework of thin-film field-effect transistors (FETs)[18] to describe the channel current $I_d$ as a function of the gate voltage $V_{gs}$ and drain voltage $V_{ds}$ of poly(3,4-ethylenedioxythiophene) polystyrene sulfonate (PEDOT:PSS)-based OECTs, according to

$$I_d = \frac{Wt}{L}\Lambda_h e \rho_0 \left(1 - \frac{V_{gs} - \frac{V_{ds}}{2}}{V_p}\right) V_{ds} \qquad (1)$$

where $V_p$ is the pinch-off voltage defined as $V_p = e\rho_0/C^*$, $\Lambda_h$ is the hole mobility (we chose this uncommon notation to avoid confusion with the chemical potential $\mu$ extensively used below), $C^*$ is the volumetric capacitance, $\rho_0$ is the intrinsic charge carrier density, $e$ is the elementary charge, and $t$, $W$, and $L$ are the geometrical channel thickness, width, and length. Bernards model is an elegant and often used framework to describe the operation OECTs. However, it does not allow for an accurate quantitative analysis of the important figures of merit of OECT operation (e.g., transconductance, saturation

[1]Laboratory for Soft Bioelectronic Interfaces, Neuro-X Institute, École Polytechnique Fédérale de Lausanne (EPFL), Geneva, Switzerland. [2]Technische Universität Dresden, Dresden, Germany. ✉e-mail: matteo.cucchi@epfl.ch

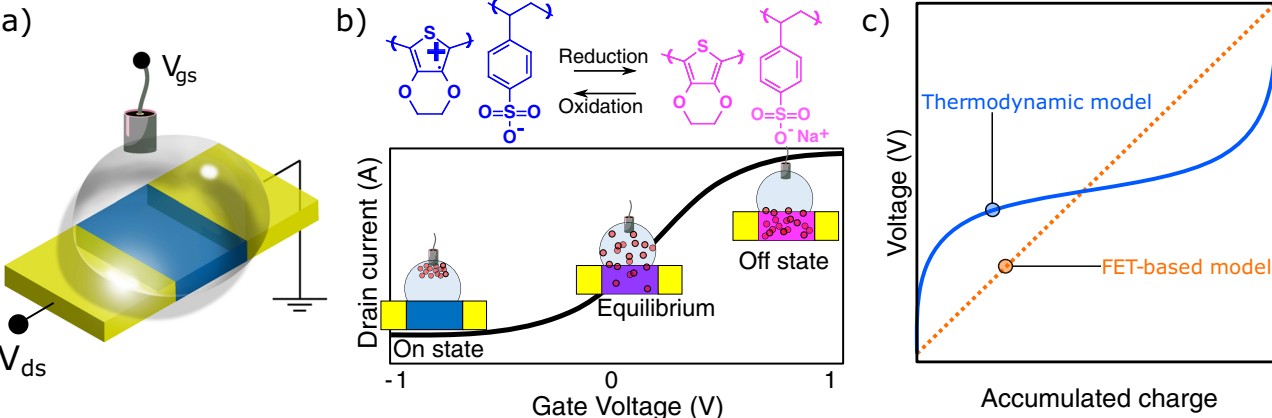

**Fig. 1 | Thermodynamic vs classic model of the OECT capacitance. a** Typical setup for the IV measurement of an OECT using a non-polarizable Ag/AgCl pellet as a gate electrode. Such setup is used throughout this work too. **b** Redox reaction driving the operation of OECTs. The amount of oxidized and reduced PEDOT leads to the transfer curve sketched below. **c** Key difference between the FET-based model for OECTs and our thermodynamic model: the nonlinear charge accumulation resulting from the Gibbs free energy, in strong contrast with a linear accumulation typical of a capacitor.

in on- and off-state, subthreshold behavior, etc.), nor for qualitative description of the OECTs IV curves, such as the peculiar S-shaped transfer curve, especially for high-performance devices. The latter, is the result of an electrochemical, rather than electrostatic, charging mechanism mediated by the redox reaction

$$PEDOT^+:PSS^- + Na^+ + e^- \rightleftharpoons PEDOT^0 + Na^+:PSS^-. \quad (2)$$

Figure 1 shows how such a reaction is reflected in the electrical properties of the PEDOT when a gate voltage is applied.

In order to address these discrepancies, several approaches have been proposed to extend Bernards model, usually adding parameters that capture OECT-specific properties not seen in conventional FETs, e.g., ion concentration-dependent mobility[19], disorder[20], or ion diffusion[21]. Bernards model and its extensions rely on the assumption that the accumulated ions in the channel can be calculated as a simple parallel-plate capacitor, similar to a metal-oxide-semiconductor FET, but experimental evidence shows a marked nonlinear potential-dependent capacitance, as shown in Supplementary Fig. 1 and ref. 22. Moreover, the chemistry behind the redox process leading to the OECT switching is not taken into account: electrostatic interactions alone cannot comprehensively describe a redox reaction and therefore cannot explain the basic mechanism that underlies the OECT operation. A few studies put forth the idea, or the necessity, to investigate the metal/electrolyte/polyelectrolyte system on a thermodynamic basis, therefore rendering the purely electrostatic picture insufficient[23]. For example, thermodynamic notions (Nernst law and Boltzmann statistics) need to be integrated into the model in order to describe the concentration-dependent threshold voltage exploited in OECTs for ion sensing[7,24]. Romele et al.[25] investigated the transport properties of OMIECs and observed that the channel resistance and capacitance are independent of the electrolyte concentration, in contrast to the double-layer theory. This ionic buffering is explained in terms of Donnan's equilibrium. Tybrandt et al.[26] reasoned in terms of the chemical potential of the charged species within the polymeric channel: their study results in accurate numerical drift-diffusion simulations and in the key observation that polyelectrolyte/conductive polymer blends should be viewed as two-phase materials. Rebetez et al. investigated the temperature-dependence of the reaction equilibrium that drives the polaron and bipolaron formation and revealed the importance of entropy in the establishment of the redox equilibrium[27]. These approaches highlight physical and chemical mechanisms behind the charging dynamics of OECTs and provide good tools to understand the interaction at the gate/electrolyte and

electrolyte/polymer interface. However, a fully-developed mathematical framework capable of reproducing experimental data while considering the material properties was never proposed. A number of empirical models and numerical simulations were also proposed[23,28,29]. A detailed overview of the mathematical models of OECTs and their evolution over time was recently reported by Colucci et al.[30].

Here, we propose a universal approach derived from thermodynamic axioms capable of explaining a plethora of OECT and OMIEC properties in a variety of applications. We argue that, although FET models can provide a useful and convenient tool to investigate the properties of OECTs, they fail in giving a realistic physical and chemical picture of the system. Instead, we reason in terms of chemical equilibrium and electrochemical potential. Key in our approach is the fact that we do not rely on the direct proportionality between applied voltage and accumulated charge typical of a parallel-plate capacitor. Rather, a nonlinear voltage-dependent charge accumulation (Fig. 1c) naturally derives from the treatment of the Gibbs energy. We analyze the equilibrium that establishes between the conductive polymer and electrolyte, and how the gate-source voltage perturbs it. In doing so, key features of OECTs naturally emerge from the math without artificial parametrization. For example, previously overlooked peculiarities of OECTs, such as saturation in the on-state and the strong voltage-dependent capacitance are intrinsically captured by our model, and so is the effect on the device performance based on the doping level.

We support our findings with experimental data employing PEDOT:PSS (the most widely employed OMIEC) OECTs as a reference system, and with numerical simulations. We capitalize on the investigation and quantification of intrinsic material properties such as the interactions occurring between the polymer and ions mediated by the solvent.

## Results

### Thermodynamics of OMIECs: redox-state equilibrium

When a PEDOT:PSS film is immersed in an aqueous electrolytic solution (e.g., NaCl in $H_2O$), the polymeric matrix swells, and ions can penetrate. Cations act as dedopants, reducing a fraction of the existing polarons $PEDOT^+$ into $PEDOT^0$ according to Eq. (2). This process happens spontaneously, without the need for external forces (e.g., a gate potential), and it is proven by the conductance lowering of PEDOT:PSS when going from salt-free solvents to the electrolytic environment. Interestingly, the resulting amount of Na:PSS is independent of the concentration of salt in the bulk electrolyte, as shown experimentally by Romele et al.[25]. This appears evident looking at the equilibrium

constant $k_{eq}$

$$k_{eq} = \frac{[PEDOT^0]}{[PEDOT^+]} \qquad (3)$$

where the activity of the electrons provided by the electrodes is considered unitary. Here, $[PEDOT^0]$ is the percentage of PEDOT that reduces after the PSS moiety pairs with a cation, and $[PEDOT^+]$ is the percentage of PEDOT in its oxidized state. Since the sum $[PEDOT^0]$ + $[PEDOT^+] = 1$ is constant, and so are the temperature and the pressure, we treat the system as a canonical ensemble with two phases[31]. Eq. (3) provides a bridge between a chemical picture of the system, since it describes the chemical reaction, and an electronic picture, because it quantifies the doping level at a certain temperature. Hence, we use it as starting point for our analysis. We set out this investigation with the definition of Gibbs free energy $G$, of a chemical reaction, and by treating the channel as a binary mixture of the two species. The definition of Gibbs energy of a binary mixture is

$$G = H - TS_{mix} = H_0 + H_{mix} - TS_{mix}, \qquad (4)$$

where $H$ is the enthalpy, $T$ is the temperature, and $S_{mix}$ is the entropy of mixing. In the simplest approximation, the enthalpy of mixing $H_{mix}$ is neglected and the only contribution to the enthalpy is $H_0$, namely the sum of the enthalpies of the pure $PEDOT^0$ and of pure $PEDOT^+$ with the corresponding chemical potentials $\mu^{p0}$ and $\mu^{p+}$, respectively[31]. By defining $\phi$ as the relative concentration of reduced PEDOT, defined as

$$\phi = [PEDOT^0] = 1 - [PEDOT^+], \qquad (5)$$

which spans from 0 (purely oxidized phase) to 1 (fully reduced phase), we can describe the entropy of mixing[31]. Accordingly, Eq. (4) becomes

$$G(\phi,T) = \underbrace{\mu^{p0}\phi + \mu^{p+}(1-\phi)}_{H_0} + \underbrace{k_B T(\phi \ln(\phi) + (1-\phi)\ln(1-\phi))}_{TS_{mix}}. \qquad (6)$$

Where $k_B$ is the Boltzmann constant. Eq. (6) is plotted in Fig. 2a for different temperatures. It is important to point out that the free energy of the whole system (electrolyte + OMIEC) should also take into account the free energy of the electrolyte $G_e$, and not only the one of the OMIEC. When ions migrate from the electrolyte into the polymer, a $\Delta G_e$ may be expected. However, such change is negligible as calculated in Supplementary Note 1. Therefore, $G_e$ is a constant, allowing us to calculate the variation of G only on the OMIEC, as well as calculate the derivative (see next section) without incurring an error.

Since the pure phases are entropically unfavorable, the systems' equilibrium at a certain temperature lies in its energetic minimum characterized by the coordinates $\phi_0$ and $G_0$. $\phi_0$ can be found by measuring the conductance of a PEDOT:PSS film in deionized water (DIW) and in saltwater. The current flowing through a film of PEDOT:PSS is given by

$$I = e\Lambda_h[PEDOT^+]\frac{Wt}{L}V = e\Lambda_h(T)[PSS^-](1-\phi(T))\frac{Wt}{L}V \qquad (7)$$

where $[PSS^-]$ is the concentration of PSS in PEDOT:PSS (in $m^{-3}$), and it is the maximum charge carrier density that the film can reach (when $\phi = 0$). This happens in DIW, where in absence of a reducing agent ($\phi = 0$) the charge density is $[PSS^-]$. Thus, by varying the temperature of the solution, the temperature dependence of the hole mobility $\Lambda_h(T)$ is found. This is shown in Fig. 2b, where a linear increase of the conductance is obtained, with a slope of 1.9 μA/K We ascribe it to a temperature-activated hopping mechanism, typical in conductive polymers. The situation is different in NaCl(aq) where both $\phi$ and $\Lambda_h$ are expected to vary with temperature. Again, a linear increase is observed. However, the slope is 1.4 μA/K, 26% lower than in DIW. By assuming $\Lambda_h(T)$ is the same for PEDOT:PSS in DIW and NaCl(aq), the difference must lay in the different charge carrier concentrations. We can therefore calculate $\phi(T)$, reported in Fig. 2c. In doing so, we are considering the hole mobility to be independent of the ion concentration. Although the mobility may depend on the doping concentration, Friedlein et al. showed that its effect is weak and should not compromise our results in the small range of dedoping level i.e., from 8% to 11.5%[19]. Accordingly, the doping level (defined as $1-\phi$) decreases, revealing that the cation concentration within PEDOT:PSS increases as a consequence of increased entropy of mixing.

A temperature-dependent doping concentration, linked to a higher entropic contribution ($\phi_0 \rightarrow 0.5$ for $T \rightarrow \infty$) is compatible with a thermodynamic picture. We now aim to generalize the mathematical framework in order to describe the behavior of PEDOT:PSS in the presence of an external gate electrode, hence, an OECT.

## Thermodynamics of OECTs: entropy of mixing
Our mathematical analysis is outlined in Fig. 3a and proceeds as follow: firstly, the equilibrium state of the reaction is analyzed (blue panel); eventually, by applying an external voltage, the equilibrium is

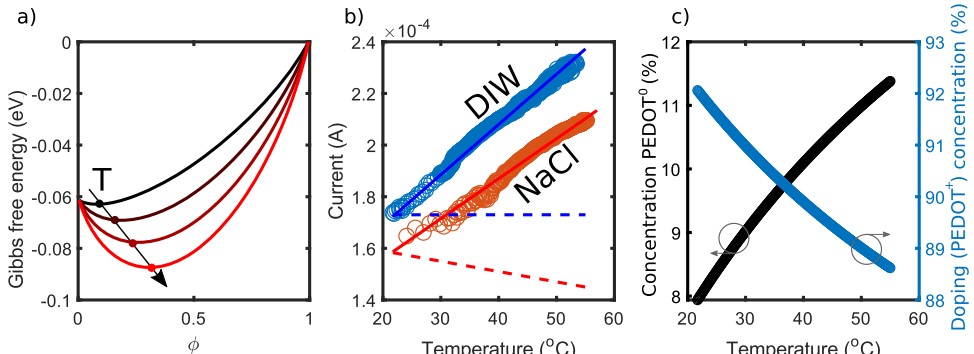

**Fig. 2 | Thermodynamics of OMIECs. a** Temperature and $\phi$ dependence of the Gibbs free energy (with $H_{mix} = 0$, using $\mu^{p+} = -60$ meV/mol and $\mu^{p0} = 0$). Higher temperatures shift the minimum towards $\phi_0 = 0.5$ and to lower energies. **b** Measurement of the current flowing in a PEDOT:PSS thin-film at a different temperatures, in deionized water (DIW) and saltwater DIW:NaCl 100 mM (circles). Curves are fitted with a linear regression (solid lines). The experiment in DIW (see Methods for details) is used to estimate the contribution of the temperature-dependent mobility, which is then subtracted, leading to the blue dashed line. The same contribution is subtracted from the current measured in DIW:NaCl. The negative slope shows that, besides the increased mobility, the doping density is lowered. Note that at room temperature, the current drops by 8% with respect to when it is in distilled water (DIW), hence $\phi_0(T = 300$ K$) = 0.08$. **c** Calculated concentration of reduced $PEDOT^0$ and doped/oxidized $PEDOT^+$ in the film at different temperatures.

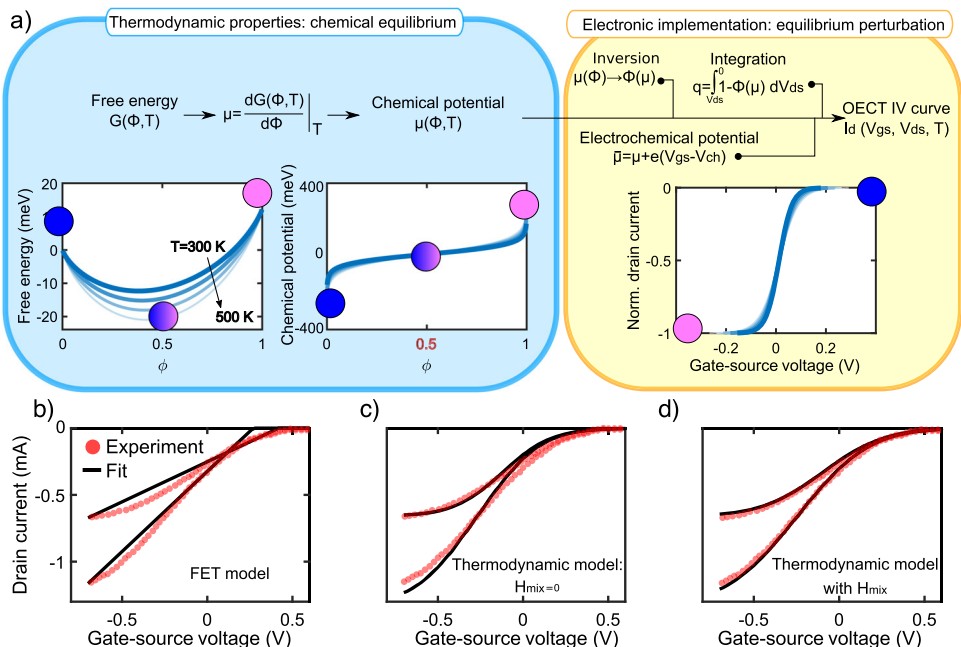

**Fig. 3 | Thermodynamic model and fits. a** Step-by-step workflow of our thermodynamic analysis. Starting with the Gibbs energy G, the chemical potential $\mu$ can be obtained. These properties are material-specific and by measuring the resistance of the PEDOT/electrolyte system only the equilibrium can be probed. By applying an external field, the equilibrium is perturbed (yellow box), and the current-voltage characteristics can be obtained as explained in the main text and in Supplementary Note 2. The pink and blue spheres indicated a fully oxidized and fully reduced PEDOT phase, respectively. **b** Transfer characteristics of a PEDOT:PSS OECT with a channel of 100 mM aqueous NaCl solution at $V_{ds} = -0.6$ V and $-0.3$ V. The experimental curves (red circles) have been fitted (solid lines) with **b** the FET-based model (Bernards model), **c** with the thermodynamic model in the assumption of $H_{mix} = 0$, and **d** with the thermodynamic model including the enthalpy of mixing. The device features $W = 50$ μm, $L = 200$ μm, $t = 200$ nm. The parameters for the fits are: $V_p = 0.57$ V and $\Lambda_h C = 40$ F/cm/s/V for the FET model. For the scenario with $H_{mix} = 0$: $\mu^{p^+} + \mu^{p0} = 1 \cdot 10^{-21} J$, $\Lambda_h [PSS^-] = 1.06 \times 10^{22}$ C/cm²/s/V, $T = 300K$, and $\alpha = 0.2$ (see Eq. Supplementary Eq. 14). When adding the enthalpy: $h_1 = 0.20$, $h_2 = 0.18$, $h_3 = 0.20$.

perturbed and the electrochemical potential must be introduced (yellow panel). the Gibbs energy can be employed, by deriving with respect to $\phi$, to obtain the chemical potential $\mu$

$$\mu(\phi,T) = \frac{dG(\phi,T)}{d\phi} = \mu^{p0} - \mu^{p+} + k_B T \ln\left(\frac{\phi}{1-\phi}\right). \qquad (8)$$

$\mu$ is plotted in Fig. 3a for different temperatures. The equilibrium state, characterized by $\mu_0 = \mu(\phi_0) = 0$, can be altered by an external stimulus e.g., an electrical (gate) potential $V_{gs}$, hence introducing the electrochemical potential in the channel $\bar{\mu}$

$$\bar{\mu} = \mu + e(V_{gs} - V_{ch}), \qquad (9)$$

with $V_{ch}$ being the potential in the channel, which depends on the drain-source voltage $V_{ds}$ according to Eq. Supplementary Eq. 4. By plugging Eq. (9) into Eq. (8), and integrating Eq. (7) over the channel length (refer to Supplementary Note 2 for the full procedure), we obtain the following expression for the current $I_d$ flowing in the OECT channel for a given $T$, $V_{ds}$, and $V_{gs}$

$$I_d = \Lambda_h \frac{Wt}{L} [PSS^-] k_B T \ln\left(\frac{Ze^{-\beta(V_{gs}-V_{ds})}+1}{Ze^{-\beta V_{gs}}+1}\right) \qquad (10)$$

where $\beta = e/k_B T$ is the ratio of the elementary charge to thermal energy and $Z = \exp(\frac{\mu^{p+}-\mu^{p0}}{k_B T})$ is a material-dependent constant that includes the chemical potential of the unmixed species. In doing so, the crucial point distinguishing this model from FET-based models is the calculation of the accumulated charge in the channel: the charge accumulation is not directly proportional to the gate-source voltage as in a plate capacitor model. Rather, it is a nonlinear function of the gate potential (see Fig. 1c) that is derived directly from the equilibrium constant in Eq. (3). Under

the condition that $H_{mix} = 0$, i.e., particle-particle interactions are neglected. While this seems a far-reaching simplification, it shines a light on some basic features of OECTs as can be seen in the output and transfer characteristics plotted in Fig. 4. Firstly, Eq. (10) includes the linear proportionality with the geometric parameters of the channel and with the hole mobility, as in the Bernards model. In contrast to it, it does not feature capacitance, underscoring the fact that we did not consider the system as a plate capacitor. Additionally, unlike traditional FETs, the equation does not need to be split between linear and saturation regimes. Rather, the dependence of $V_d$ becomes weaker as $V_g$ becomes more negative, and the saturation at large drain voltages is inherently captured. Moreover, it reproduces the saturation in the off-state (due to the complete PEDOT reduction ($\phi = 1$) as well as the saturation in the on-state (complete oxidation ($\phi = 0$)) as shown in Fig. 3a. The latter was never described in previous theories for OECTs. From a thermodynamic standpoint, $\mu(\phi = 0) \rightarrow -\infty$ and $\mu(\phi = 1) \rightarrow +\infty$. This means that huge changes in chemical potential i.e., large variation of $V_{gs}$, bring about small changes in cation concentration, hence saturation of the drain current. Finally, this model confirms a shift of the threshold voltage at different temperatures[32] and foresees that, at room temperature, a gate voltage of <1 V is needed to drive a PEDOT:PSS-based OECT from the on-state to the off-state, in good agreement with state-of-the-art OECTs. In Fig. 3b, c, the transfer characteristics are fitted with the thermodynamic model and compared to the traditional FET-based model (Bernards model), revealing a much better agreement with experimental data even for the simplistic model without particle-particle interactions. In Fig. 2 the same curves are reported on a logarithmic scale. Nevertheless, we stress that this model does not aim to describe the properties of the channel for currents below the threshold voltage, where the leakage current becomes non-negligible and diffusion currents may play an important role. Moreover, decoupling thermodynamic and kinetic effects is a challenging task.

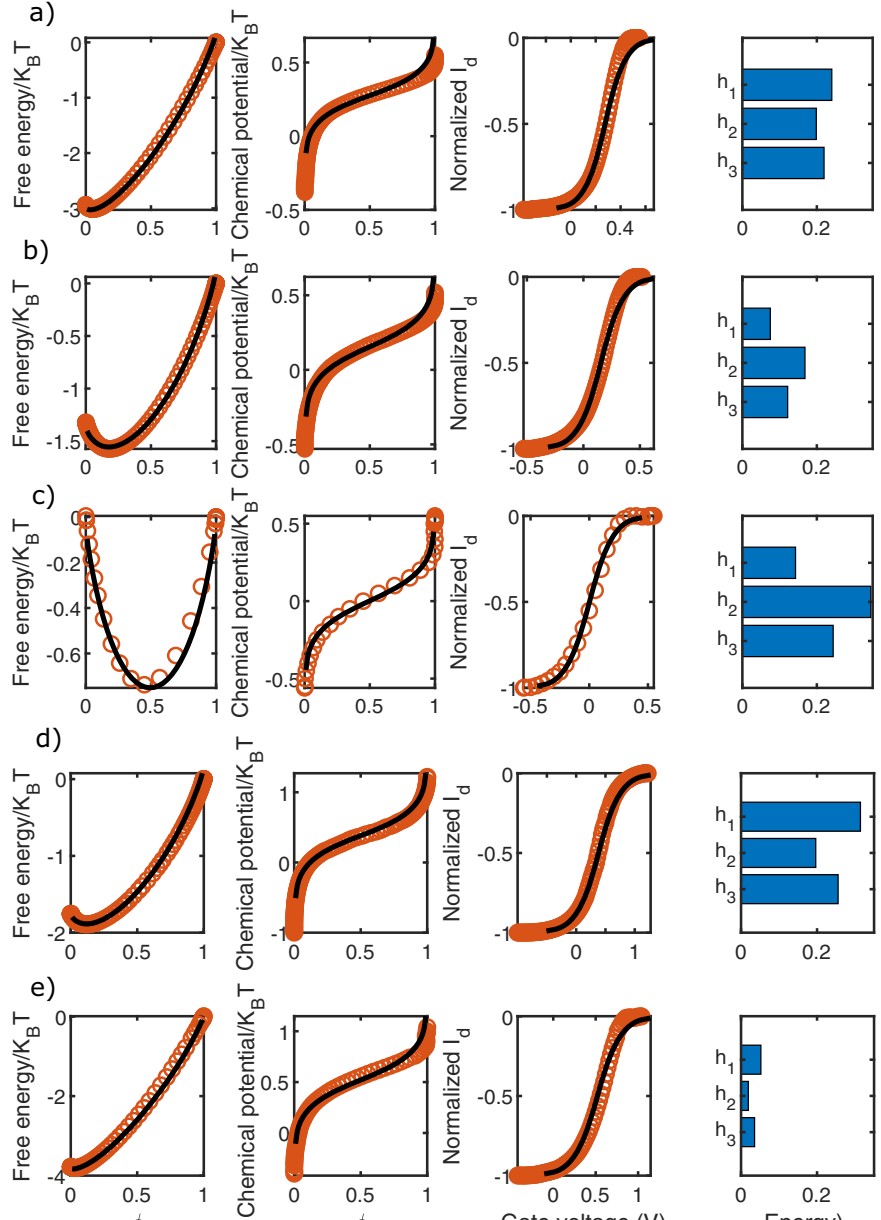

**Fig. 4 | Quantification of enthalpy.** Investigation of the chemical potential and free energy underlying the transfer characteristics of an OECT immersed in **a** 100 mM NaCl aqueous solution, **b** 100 mM KCl aqueous solution, **c** ionic liquid, **d** 100 mM LiClO$_4$ in butanol, and **e** 100 mM LiClO$_4$ in water. Experimentally determined curves with $V_{ds}$ = 5 mV (circles) are fitted with the blue solid curves. The histograms on the right display the parameters used for the fits. In Supplementary Table 1 the values and the errors of the parameters used for the fits are reported, together with the fitting procedure. All energies are in $k_B T$ units.

To this end, each voltage sweep was performed very slowly (1 V/min) to ensure that the system is always at equilibrium. This is a "static" model, and the description of transient effects may go beyond the thermodynamic approach.

**Enthalphy of mixing**

In order to achieve better quantitative agreement with experimental data, we must go beyond the interaction-free approximation and allow for enthalpic contributions $H_{mix}$ arising from the interaction between reduced PEDOT pairs (interaction parameter $h_1$), between oxidized PEDOT pairs (interaction parameter $h_2$), or between each other (interaction parameter $h_3$) using the following term describing interactions as

$$H_{mix} = k_B T(h_1 \phi^2 + h_2(1-\phi)^2 + h_3 \phi(1-\phi)). \tag{11}$$

Differently from the previous scenario, $\phi(\mu)$ cannot be explicated if $H_{mix} \neq 0$. We need to resort to numerical solutions for the integration in Eq. Supplementary Eq. 7 plotting and data fitting (see Supplementary Note 3 for the full code). With this method, we fit the transfer characteristics as shown in Fig. 3d: the agreement with experimental data is striking and decisively improving with respect to the scenario with $H_{mix} = 0$.

The presence of underlying S-shaped chemical potential can be directly probed by measuring a transfer curve for $V_{ds} \rightarrow 0$ (See Eq. Supplementary Eq. 10). Lowering the drain-source voltage to such a low value (5 mV in our experiments) makes the channel potential homogeneous and, mathematically speaking, compresses the integral along the channel length to a point (See Supplementary Fig. 3). We do this for a number of material combinations, fit the data, and analyze the parameters used for the modeling.

For example, using larger cations (K instead of Na as in Fig. 4a, b), the charges are forced to stay farther away, lowering the Coulomb interaction energy (which we believe is mainly reflected in the decrease of the interaction parameters $h_1$, $h_2$, and $h_3$). Moreover, different solvents can be used to validate the model. If a water-free solvent is used, the enthalpy of mixing is larger than in the case of water. This is visible when employing an ionic liquid (Fig. 4c) or butanol (Fig. 4d) as electrolyte. Figure 4d, e compare OECTs immersed in a solution of 100 mM LiClO$_4$, first in butanol and then in water. Water, with its higher dielectric constant, significantly lowers the contribution of $H_{mix}$, again reflected in the decrease of the interaction parameters.

Besides the trend, it is worth discussing the absolute values of the $H_{mix}$ contribution: its magnitude is a small correction to the interaction-free scenario with $H_{mix} = 0$. This may be surprising as the electrostatic interaction that arises between holes, cations, and polyelectrolyte may make a larger effect be expected. This apparent paradox is readily resolved by reasoning in terms of energy variation $\Delta G$: in the fully oxidized phase, the PSS dopants are paired with positively charged polarons. When cations are injected, they replace the polaronic charge without altering dramatically the energetic landscape. This process is sketched in Supplementary Fig. 5. Therefore, the driving force behind the operation of an OECT is entropy, with small perturbations coming from the enthalpic terms. Therefore, for targeted material design, the entropic term must be addressed in order to change the device's figures of merits decisively.

### Implications for device and material design

The thermodynamic model offers a way to understand OMIECs and OMIEC-based devices, as well as a pathway for intelligent device engineering and material design. The very fact that chemical equilibrium is at the core of the operation of an OECT (i.e., the system lies at the minimum of the Gibbs energy function), leads to a transconductance that is highest for $V_{gs} = 0$, which is an excellent feature for possible self-powered devices. Note that this is true only if the drain voltage approaches zero, and there may be an additional shift due to the reaction at the gate/electrolyte interface. Therefore, minimizing the channel length ensures a high on-state current despite the low $V_{ds}$ used. Moreover, the same approach (low $V_{ds}$, small L), allows for an easier mathematical analysis of the entropic e enthalpic analysis and fitting and should be used to compare different OECTs.

As discussed above, the entropic term dominates the enthalpic, giving the OECTs a dissimilar behavior when compared to traditional FETs and OFETs. Unfortunately, the entropic term is also very challenging to tune, posing the difficult question of how to improve the device's performance. On the contrary, one can change the enthalpic term more easily by employing different ions and solvents, although these may be dictated by the application. In general, by decreasing the enthalpic contribution $h_1$ and $h_2$, as well as increasing $h_3$, the resulting transconductance of the device can be increased. This can be seen by solving numerically Eq. (4).

Another point to pay attention to is the type of material used. Our measurements have been carried out using PEDOT:PSS channels. We believe, however, that the model it can be extended to any type of depletion mode OECTs in which a scheme analogous to the one in Supplementary Fig. 5 applies. However, novel OMIECs allow for accumulation mode OECTs, where the species in the off-state may be fully neutral (i.e., only the neutral polymer without any dopant), and only in the on-state ions generate charge carriers. On the one hand, the model cannot be readily interpreted for accumulation mode OECTs. On the other hand, the theory is not material-specific but rather universal. The basic features deriving from the thermodynamic model e.g. S-shaped transfer curves, bell-shaped transconductance, and transition from on- to off-state in the range of 1 V[33,34] still apply to such devices, suggesting that the model may be applied to them as well. In doing so, one must adjust the initial equations and take into account the fact

that the device may be off-state when is doped (See Supplementary Eq. 11).

## Discussion

A thermodynamic analysis of the OECTs is presented. Our approach stands on general thermodynamic axioms and leads to a comprehensive, compact, and quantitative framework capable of describing the current-voltage characteristics of such devices and how these arise from the interactions of the material. We reveal that the main driver in the switching of the OECT is the entropy of mixing, in strong contrast with traditional models based on purely electrostatic effects. By measuring the channel current at very low drain voltages, we are able to quantify the enthalpic term of the total free energy, and conclude that it only represents a minor contribution when compared to the entropy (1–10% depending on the system). We also highlight that some important advantages of OECTs, e.g., high transconductance and low voltage operation, are purely a consequence of such thermodynamic features. Our model allows excellent fitting of experimental transfer curve and leads to the conclusion that a model based purely on electrostatic interactions, like the ones typically used, is insufficient to interpret experimental data correctly and unravel chemical and physical interaction of the material with the ions. In fact, the electrostatic (enthalpic) contribution is minor and we quantified it across a large range of materials. Our approach can be used for a better understanding of the physical and chemical interactions that drive the operation of OECTs, as well as a precise guideline for targeted material modeling and engineering.

## Methods

OECTs were fabricated on $1 \times 1$ inch glass substrates covered with Cr (3 nm) and Au (50 nm). The photoresist AZ 1518 (MicroChemicals GmbH) was spin-coated (3000 rpm for 60 s; SAWATEC AG), followed by baking at 110 °C for 60 s. Gold traces were shaped by illuminating in a mask aligner photolithography system (I-line 365 nm, lamp power 167 W, SUESS Microtec AG) for 10 s, photoresist developing (AZ 726 MIF, MicroChemicals GmbH), and Au- and Cr-etching for 60 s and 20 s (10% diluted aqueous solutions of Standard Gold/Chromium etchants, Merck KGaA). After O$_2$-plasma cleaning, PEDOT:PSS (Clevios PH1000, Heraeus Deutschland GmbH & Co. KG) with 5% v/v ethylene glycol (Merck KGaA) was spin-coated (2000 rpm for 60 s) to yield an ~200 nm thick layer. This step was followed by drying at 120 °C for 20 min. Orthogonal photoresist OSCoR 5001 (Orthogonal Inc.) was spin-coated (3000 rpm for 60 s), baked for 60 s at 100 °C, and exposed for 12 s to structure channel and gate. After post-baking (60 s at 100 °C), development followed by covering the sample with Orthogonal Developer 103a (Orthogonal Inc.) and removing it by spinning after 60 s (twice). Excess PEDOT:PSS was removed by O$_2$-plasma etching (5 min, 0.3 mbar, Diener electronic GmbH & Co. KG), after which the sample was placed in Orthogonal Stripper 900 (Orthogonal Inc.) overnight at room temperature. Electrical characterizations were performed using Keithley 2600 SMUs controlled by the software SweepMe! (sweep-me.net) with the PEDOT:PSS film immersed in 100 mM water:NaCl. The transfer curves were measured slowly (1 V/min) using an Ag/AgCl pellet as a gate in order to ensure the system operated at equilibrium and so minimized transient and hysteretic behaviors. After a preliminary test to establish where the saturation in the on and off states occurred, the measurement was carried out from the on-state to the off-state. The temperature-resolved measurement was carried out with the device immersed in a beaker filled with water and NaCl 100 mM, with the beaker on top of a hotplate, and a thermocouple attached to the substrate. A voltage of 10 mV was applied constantly, while temperature and current were controlled with a parameter analyzer Keythley interfaced with the software SweepMe! (sweep-me.net). Numerical simulations were coded in Matlab, code in Supplementary Note 3. In Fig. 4, The experimental I–V transfer curves

are transformed into a $\mu - \phi$ as explained in Supplementary Note 2. This is possible only because the curves are measured at vanishing drain voltage. By integrating the chemical potential, the Gibbs free energy is obtained, which is the curve on which the fit is carried out. In order to measure a significant current even with very small Vd (5 mV), the channel was patterned to be wide and short: $L = 50\ \mu m$, $W = 500\ \mu m$.

## Data availability

Data available on request from the authors or downloadable from https://osf.io/dufsz/?view_only=e884dff34dc646599f95ae02a44d42ce.

## Code availability

Codes are included in the Supplementary Information (Note S3). For more information please contact the corresponding author.

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

## Acknowledgements

We thank Dr. Ron Dockhorn for the fruitful conversations about the thermodynamics of conductive polymers. H.T. thanks the financial support from the European Social Fund (ESF) and the Free State of Saxony in the framework of OrgNanoMorph (project number: 100382168). L.M.B. and H.K. are grateful for funding from the German Research Foundation (DFG, KL 2961/5-1). A.W. thanks the project ESF Re-Learning (cfaed), ESF Number: 100382146.

## Author contributions

M.C. originated the idea, developed the math, and designed the experiments. A.W., L.M.B., H.T., H.K., and K.L. equally contributed to the discussion. R.K. generated the code for the fits for the enthalpy of mixing.

## Funding

## Competing interests

The authors declare no competing interests.
