## [Peer Review File · Nature Communications]

Thermodynamics of Organic Electrochemical TransistorsREVIEWER COMMENTS

Reviewer #1 (Remarks to the Author):

Referee comments

The manuscript reports a development of the description of OECT devices with OMIEC materials, that goes beyond the engineering approach which uses the description of an equivalent capacitance. Here, a thermodynamic treatment is given that appears to give a superior matching to experimental data. I find the ambition very interesting, and could maybe even be convinced of the relevance, were it not for a number of weaknesses and ambiguities in the presentation. These are severe enough that I consider the manuscript to need an extensive revision. Still, I do think that the authors may have made an important step to improve the still rather primitive analysis of OECT devices, and encourage a more transparent presentation of their model, taking account of some of the criticisms raised below.

I have one overriding consideration: When analysing the thermodynamics of the OECT channel, the free energy of the surrounding electrolyte is not included; only that of the OECT channel. But without the ion exchange between these two different materials phases, the modulation of channel conductance would not happen. I do not understand how both of these elements must be integrated in a full model, but I do not agree that one of them can be neglected.

As I consider that the entropy due to ions moving between active channel and the adjacent electrolyte must be included in the treatment, I also suggest that the identification of the enthalpy is dubious, as it is formed by the difference (which will include the entropic elements from the electrolyte, if these are sizable). Thus the assignments of enthalpy for different electrolytes may need revision.

Equation (1) specifies that the density of dopant ions is equivalent to the density of charge carriers, thus specifying that all dopant ions are paired with a hole. There are other material systems in OFETs and OECTs where this identity is not necessary; and while it may be true in the case of the PEDOT(PSS) studied here, there is always the chance that a sulphonate ion is paired with a proton/cation, rather than a hole. I suggest a more general terminology, or maybe just a clarity in assigning symbols.

The formula is also missing the elementary charge, making it dimensionally incorrect. That is better handled in the supplemental.

On page 3, it is stated that the conductance of a PEDOT(PSS) film is lower when immersed in salt free solvents than when immersed in the with solvents with salt. Having worked with PEDOT(PSS) for 30 years, this was an astounding and novel statement for me, and I have tried in vain to find earlier evidence for this statement, from experiences collected during my life as an experimentalist. I had to resort to experiments. These were very primitive and with many limitations, but they did not show me that significant modulation of conductivity is obtained. I am not considering these results significant, but rather note that more controlled and extensive experiments will be necessary to verify the statement.

Even if my experiments were primitive and non-decisive, I would ask for more definitive experiments showing that the introduction of a salt into the solvent leads to increased conductance in a

PEDOT(PSS) film. In the Fig 2b where such experiments are reported, the description does not even include a concentration for the NaCl solution. The description of experiments needs improvement.

On page 4 it is stated that

I believe that what is intended is quite the opposite to what is stated, and do not quite understand the text; I do understand that "the magnitude" refers to the concentration dependence of the hole mobility. The existing effect is thus the one that is negligible? This could be better expressed. And referencing to Friedlein is also more clear if using references---there are at least two of them in the reference list, which I had to browse. Not the task of readers, I think.

ON page 7, it is stated that. ---

and I do not understand what "stay further" is?

Reviewer #2 (Remarks to the Author):

The manuscript "Disentangling the Entropic and Enthalpic Contributions in the Operation of the Organic Electrochemical Transistor" presents a thermodynamic framework for explaining the current-voltage characteristics of OECTs from the treatment of Gibbs free energy. While the paper is well written and presents interesting results, there are scopes for improvement. In the current form, the paper is not recommended for publication unless following changes are made.

1. In Figure 1b, the direction of oxidation and reduction seems been flipped.
2. Page 5, last paragraph. Figure 3c is incorrectly labeled as 3e.
3. Figure 2b. Specify the device dimensions for this measurement. What was the voltage applied? Update Methods.
4. In page 7, paragraph 1, authors give the notion that ϕ_0 occurs at $V_{GS}=0V$. But this is not true. When a gate electrode is connected, the electric field between gate-drain cannot be neglected even at $V_{GS}=0V$, and therefore affects the equilibrium doping level of the channel if V_{DS} is sufficiently high. Therefore, ϕ_0 must be the equilibrium achieved by passive diffusion of ions without connecting a gate electrode. Besides, occurrence of $g_{m,max}$ at $V_{GS}=0 V$ is a special condition, and not universally applicable for all the materials used as OECT channel, including various combinations of PEDOT:PSS and its dopants. It also related to the channel dimension owing to the relative potential drops at gate and channel. Authors must present the data (I vs t @ $V_{GS}=0V$) for change in channel current before and after physically removing the gate electrode.
5. All figures presenting transfer curves should show the data in log scale for a clear picture.
6. Figure 4a-e. What is the magnitude of parameters (free energy, chemical potential, and drain current) plotted in the graphs? What is the fitting error for each curve?

7. How does the proposed equation capture changes in device IV characteristics in relation to the changes in morphology? For example, a PEDOT:PSS-based OECT IV characteristics before and after an annealing step?

8. The equations fit well for the depletion mode OECT presented in the work. However, their generalizability is not demonstrated to other materials, including rising high performance materials such as accumulation mode materials. How does equation 7 change for accumulation mode materials? How does the expression S8 change for polymers that are not doped to begin with? In its current form, the manuscript does not prove its universality.

9. In Figure 3b, the linear I-V relationship in FET model is applicable in the linear region, how about the case in the saturation region? The device should not operate in the linear region considering the large VDS the authors use.

10. In the last paragraph of page 5, the authors say, "This means that huge changes in chemical potential i.e., large variation of V_{gs} , bring about small changes in cation concentration, hence saturation of the drain current". Can the authors substantiate this using data? For example, using a current controlled mode (variable/constant I_{gs} , and measuring VGS, IDS)?

11. In page 7, "...in the fully reduced phase, the PSS dopants are paired with positively charged polaron." this statement is incorrect, it should be in the fully oxidized phase.

12. What was the scan rate used for collecting IV curves? What was the scanning direction? Do the authors observe hysteresis? What was the leakage current during measurements? Please include these details in the appropriate sections.

13. Methods. More details about the device fabrication must be provided. Were the electrodes patterned through lift-off or RIE etching? How was the channel patterned (RIE)? Was there a post-annealing step? What SMU was used for the measurements? How was the volumetric capacitance measured? Figure 4a-e shows data for OECTs using liquid electrolytes. But in methods, measurements using solid electrolytes is also mentioned.

14. Although the entropic contributions seem to well describe steady state behavior of the presented OECT, the manuscript fails to communicate its transient behavior. What are the limitations of the current approach? How does the method compare against the approximations put forward by ref 25 (fig 2b,c,f)?

15. The entropic term is mainly determined by ϕ . The manuscript doesn't clearly suggest a trajectory for improving the device performance. In light of the presented results, how can the materials be tailored for different applications? For example, high gain vs high speed? How to use this framework to compare among materials based on the proposed model?

REVIEWER COMMENTS

Reviewer #1 (Remarks to the Author):

Referee comments

The manuscript reports a development of the description of OECT devices with OMIEC materials, that goes beyond the engineering approach which uses the description of an equivalent capacitance. Here, a thermodynamic treatment is given that appears to give a superior matching to experimental data. I find the ambition very interesting, and could maybe even be convinced of the relevance, were it not for a number of weaknesses and ambiguities in the presentation. These are severe enough that I consider the manuscript to need an extensive revision. Still, I do think that the authors may have made an important step to improve the still rather primitive analysis of OECT devices, and encourage a more transparent presentation of their model, taking account of some of the criticisms raised below.

We thank the reviewer for taking the time to carefully read the manuscript. We believe you raised excellent points and after addressing them, the work gained substantial value. Please find below our comments/responses (in red), as well as the modifications we applied to the main text (*in italic*). The same modifications can be found in the pdf highlighted in yellow.

I have one overriding consideration: When analysing the thermodynamics of the OECT channel, the free energy of the surrounding electrolyte is not included; only that of the OECT channel. But without the ion exchange between these two different materials phases, the modulation of channel conductance would not happen. I do not understand how both of these elements must be integrated in a full model, but I do not agree that one of them can be neglected.

Excellent point. We did not think about it from the start, and indeed it makes a difference between a correct theory and a wrong one. However, after careful analysis, we believe our procedure is still valid. We reason as follows:

The free energy of the surroundings depends on the temperature and the ion concentration. If a considerable number of ions were to move from the electrolyte to the film, the free energy of the electrolyte would change. However, the beaker in which the channel is immersed is immensely larger than the channel itself. This can be quantified: the number of ions that move from the electrolyte to the channel is roughly 10% of the PSS contained in PEDOT:PSS (Fig. 2b). If the size of the channel is $200\ \mu\text{m} \times 50\ \mu\text{m} \times 100\ \text{nm}$, and the concentration of PSS is $7.4 \times 10^{14}\ \text{cm}^{-3}$ (Journal of Applied Physics 114, 133708 (2013), Rutledge et al.), one finds that the amount of ions that go in PEDOT:PSS is roughly $7.4\text{E}4$ ions.

In comparison, the electrolyte (beaker as used in the experiments) contains an amount of ions in the order of the Avogadro's number. The change of entropy that accompany this is given by

$$\Delta G = RT \log(c_1/c_2)$$

where c_1 is the concentration of the electrolyte before entering in contact with PEDOT:PSS, and c_2 is the one after. But as we calculated above, c_1 and c_2 are basically the same, giving a $\Delta G = 0$.

Similarly, the enthalpy depends on the interaction between ions, and therefore on their ability of coming close, hence, again on the concentration. Therefore, the thermodynamics of the electrolyte cannot be altered by the presence of such a small channel.

We conclude that the Gibbs free energy of the electrolyte should be counted, but it adds up to the free energy of the polymeric system as a constant. As such, when measuring variation of energy (ΔG), or when we do the derivative to calculate the chemical potential, such a constant is lost and does not affect our theory.

We now added a new Supplementary note 1, where we address these calculations in more detail. Furthermore, we highlight in the supplementary note 1 that the contribution of the electrolyte needs to be added to the theory if systems a small number of ions are investigated (e.g., a layer of PEDOT with a thin printed solid-state electrode

As I consider that the entropy due to ions moving between active channel and the adjacent electrolyte must be included in the treatment, I also suggest that the identification of the enthalpy is dubious, as it is formed by the difference (which will include the entropic elements from the electrolyte, if these are sizable). Thus the assignments of enthalpy for different electrolytes may need revision.

As discussed above, and in Supplementary note S1, we believe the point is correct, but being a constant, it does not harm the foundations of the model based on derivatives. However, it might become an important effect for patterned solid electrolytes where the electrolyte film is comparable in size to the PEDOT:PSS film.

We also specify in the main text:

It is important to point out that the free energy of the whole system (electrolyte + OMIEC) should also take into account the free energy of the electrolyte G_e , and not only the one of the OMIEC. When ions migrate from the electrolyte into the polymer, a ΔG_e may be expected. However, such change is negligible as calculated in Supplementary Note 1. Therefore, G_e is a constant, allowing us to calculate variation of G only on the OMIEC, as well as calculating the derivative (see next section) without incurring in error.

Equation (1) specifies that the density of dopant ions is equivalent to the density of charge carriers, thus specifying that all dopant ions are paired with a hole. There are other material systems in OFETs and OECTs where this identity is not necessary; and while it may be true in the case of the PEDOT(PSS) studied here, there is always the chance that a sulphonate ion is paired with a proton/cation, rather than a hole. I suggest a more general terminology, or maybe just a clarity in assigning symbols.

Thank you for this observation. We absolutely agree with you and now use the general "charge carrier density ρ in the revised manuscript.

The formula is also missing the elementary charge, making it dimensionally incorrect.

That is better handled in the supplemental.
Thank you for the remark. It has been now fixed

On page 3, it is stated that the conductance of a PEDOT(PSS) film is lower when immersed in salt free solvents than when immersed in the with solvent with salt. There seem sto be a misunderstanding: the text tries to express the opposite. The conductance is lower when immersed in the solvent with salt: "This process happens spontaneously, without the need of external forces (e.g., a gate potential), and it is proven by the conductance lowering of PEDOT:PSS when going from salt-free solvents to electrolytic environment".

Having worked with PEDOT(PSS) for 30 years, this was an astounding and novel statement for me, and I have tried in vain to find earlier evidence for this statement, from experiences collected during my life as an experimentalist. I had to resort to experiments. These were very primitive and with many limitations, but they did not show me that significant modulation of conductivity is obtained. I am not considering these results significant, but rather note that more controlled and extensive experiments will be necessary to verify the statement.

We repeated this measurement for different samples and geometries and we are convinced that about this finding. We agree that the change in resistance is not so dramatic as you say (always about 8-10% lower at room temperature as shown in Fig. 2b). To notice the change in DI water, one must wash away thoroughly the excess ions that may come from the precursor materials or from previous measurements. This observation is not too surprising to us considering that some ions may mediate dedoping even without gate voltage.

Even if my experiments were primitive and non-decisive, I would ask for more definitive experiments showing that the introduction of a salt into the solvent leads to increased conductance in a PEDOT(PSS) film. In the Fig 2b where such experiments are reported, the description does not even include a concentration for the NaCl solution. The description of experiments needs improvement.

Thank you for the remark. In the revised manuscript we added the information in the caption (100 mM, now added to the caption). Furthermore, the text stresses the fact that Romele et al (ref. 24) showed that the concentration is not important, in agreement with our model based on chemical equilibrium (figure below).

On page 4 it is stated that

I believe that what is intended is quite the opposite to what is stated, and do not quite understand the text; I do understand that “the magnitude” refers to the concentration dependence of the hole mobility. The existing effect is thus the one that is negligible? This could be better expressed.

Thank you. As it was written it may have been misleading. Yes, we refer to the hole mobility and the effect that charge/doping concentration has on it. We edited the text as follow:

Although the mobility may depend on the doping concentration, Friedlein et al. showed that its effect is weak and should not compromise our results in the small range of dedoping level i.e., from 8% to 11.5% [ref 18, friedelin et al 201].

And referencing to Friedlein is also more clear if using references---there are at least two of them in the reference list, which I had to browse. Not the task of readers, I think. Thank you for the comment, this has been fixed.

ON page 7, it is stated that. ---
Sorry, this part is missing...

and I do not understand what “stay further” is?
to stay farther, at larger distance. PSS and the large cations are more distant because of their size.
We now changed “further” to “farther away”.

Finally, we believe a more succinct title such as “thermodynamics of organic electrochemical transistors” is more clear and appropriate

Reviewer #2 (Remarks to the Author):

The manuscript "Disentangling the Entropic and Enthalpic Contributions in the Operation of the Organic Electrochemical Transistor" presents a thermodynamic framework for explaining the current-voltage characteristics of OECTs from the treatment of Gibbs free energy. While the paper is well written and presents interesting results, there are scopes for improvement.

In the current form, the paper is not recommended for publication unless following changes are made.

We thank you for the careful reading and positive feedback, and for taking the time to raise good points that resulted in a better and much comprehensible manuscript. Please find below our comments/responses (in red), as well as the modifications we applied to the main text (*in italic*). The same modifications can be found in the main text highlighted in yellow. We answered all your points and, as a result of your observation, strongly improved the Methods sections and the added a final section to better put forth our results in a broader perspective.

1. In Figure 1b, the direction of oxidation and reduction seems been flipped.

Thank you for pointing out this mistake! We fixed it.

2. Page 5, last paragraph. Figure 3c is incorrectly labeled as 3e.

Thank you. Actually, it should refer to 3b. now fixed.

3. Figure 2b. Specify the device dimensions for this measurement. What was the voltage applied?

As for the caption in Fig. 2b, it does specify both the voltages used and the dimension of the channels. More details in the Methods sections.

Update Methods.

Fair point. We have now greatly extended and refined the Methods section, adding details about measurements and fabrication.

4. In page 7, paragraph 1, authors give the notion that ϕ_0 occurs at $V_{GS}=0V$. But this is not true. When a gate electrode is connected, the electric field between gate-drain cannot be neglected even at $V_{GS}=0V$, and therefore affects the equilibrium doping level of the channel if V_{DS} is sufficiently high. Therefore, ϕ_0 must be the equilibrium achieved by passive diffusion of ions without connecting a gate electrode.

We agree with your observation, which is in general true. Therefore, all these measurements are carried out using a v_d that is exceedingly small ($V_d=5mV$). In this case, the channel current at $V_g=0$ or absent V_g should be the same. We confirmed with an experiment:

We specify that this is a good assumption only in this case. Please refer to Supplementary Fig. 2, Supplementary Note 2 and Eq. S10

Besides, occurrence of $g_{m,max}$ at $V_{GS}=0$ V is a special condition, and not universally applicable for all the materials used as OECT channel, including various combinations of PEDOT:PSS and its dopants. It also related to the channel dimension owing to the relative potential drops at gate and channel.

Excellent point. We removed from the main text all the claims referring to $g_{m,max}$ at $V_g=0$. In Supp. Note 2 we report the calculation of g_m and its dependencies on V_g and V_d .

Authors must present the data (I vs t @ $V_{GS}=0V$) for change in channel current before and after physically removing the gate electrode.

Please refer to the figures attached above and the following time-dependent measurement.

The current difference is very small. It is of course an assumption, but a good one under the condition that V_d is very small

5. All figures presenting transfer curves should show the data in log scale for a clear picture.

We purposefully avoided introducing figures in log scale, and if you agree we would stick to it. The reason is that the model does not cover subtleties happening in the off state,

therefore it does not - and we do not claim - that it can reproduce the property of the transistor below the threshold. This is a feature common to most of the models used to describe FETs and MOSFETs and to the best of our knowledge no model can describe both regimes accurately. In the sub-threshold regime, the leakage current becomes non-negligible and/or dominant and diffusion currents plays an important role. We have now noted this in the main text in Sec. 3

We stress that this model does not aim to describe the properties of the channel for currents below the threshold voltage, where the leakage current becomes non-negligible and diffusion currents may play an important role.

Please find attached the figure 3bcd in log scale showing the leakage. But we insist, if you agree, that it should not be included in order to avoid confusion for the reader

6. Figure 4a-e. What is the magnitude of parameters (free energy, chemical potential, and drain current) plotted in the graphs?

Thanks for pointing it out. Units are adimensional (Energy/KbT), now added to the y label.

What is the fitting error for each curve?

We added table S1 with all the fit values and their errors

7. How does the proposed equation capture changes in device IV characteristics in relation to the changes in morphology? For example, a PEDOT:PSS-based OECT IV characteristics before and after an annealing step?

We expect the orientation of the polymeric chains to change during annealing which improves the hopping mobility of the holes (together with the stability of pedot). As this effect can be observed even in dry PEDOT, it does not necessarily relate to the thermodynamics of the ions in solution. Our model capture this as the mobility dependence is included in all the equations (Λ_h)

In Sec. 3 we say:

Eq. 10 includes the linear proportionality with the geometric parameters of the channel and with the hole mobility, as in the Bernards model.

8. The equations fit well for the depletion mode OECT presented in the work. However, their generalizability is not demonstrated to other materials, including rising high performance materials such as accumulation mode materials. How does equation 7 change for accumulation mode materials?

Very good point. We now elaborate this in a whole new short section 5 right before the conclusions named "implications for device and material design" which condenses how our findings can be translated to other materials and device design. In this section, we now mention our conclusions on accumulation mode OECTs, leaving the door open to future work.

Another point to pay attention to is the type of material used. Our measurements have been carried out using PEDOT:PSS channels. We believe, however, the model it can be extended to any type of depletion mode OECTs in which a scheme analogous to the one in Supplementary Fig 4 applies. However, novel OMIECs allow for accumulation mode OECTs, where the species in the off-state may be fully neutral (i.e. only the neutral polymer without any dopant), and only in the on-state ions generate charge carriers. On the one hand, the model cannot be readily interpreted for accumulation mode OECTs. On the other hand, the basic features deriving from the thermodynamic model e.g. S-shaped transfer curves, bell-shaped transconductance, and transition from on- to off-state in the range of 1 V still apply to such devices, suggesting that the model may be applied to them as well. In doing so, one must adjust the initial equations and take into account the fact that the device may be in off-state when is doped (See Eq s11).

How does the expression S8 change for polymers that are not doped to begin with? In its current form, the manuscript does not prove its universality. Also discussed in Section 5 now.

9. In Figure 3b, the linear I-V relationship in FET model is applicable in the linear region, how about the case in the saturation region? The device should not operate in the linear region considering the large VDS the authors use.

In the examples in figure 3b, the device should still operate in the linear regime $|V_g - V_{th}| < V_d$. The transition from linear to saturation regime in OECTs is generally less marked than in FETs and concepts such as the pinch-off are not as solid in OECTs. Moreover, even using a quadratic expression as in the case of the saturation regimen in FETs will not result in a good fit given the S-shaped transfer curve. Looking at the equation for the drain current, for large v_d , the dependency on V_g becomes weaker. We have now added a Fig. S3, showing output and transfer curves for high v_d , containing important comments on their shape.

10. In the last paragraph of page 5, the authors say, "This means that huge changes in chemical potential i.e., large variation of V_g s, bring about small changes in cation concentration, hence saturation of the drain current". Can the authors substantiate this

using data? For example, using a current controlled mode (variable/constant I_{gs} , and measuring VGS, IDS)?

This measurement was unsuccessful. While keeping a fixed I_{gs} , one measures the leakage (gate) current e.g. the current from the contacts or capacitive current, but we do not gain any information on the channel accumulation. However, we insist on the very definition of "saturation", where the current reaches a plateau and the derivative over V_g tends to zero. In general, it is very difficult to operate an OECT with a constant gate current.

11. In page 7, "...in the fully reduced phase, the PSS dopants are paired with positively charged polarons." this statement is incorrect, it should be in the fully oxidized phase.

Thank you. We changed to "*fully oxidized*".

12. What was the scan rate used for collecting IV curves? What was the scanning direction? Do the authors observe hysteresis? What was the leakage current during measurements? Please include these details in the appropriate sections.

The scan rate was extremely slow (1 V per minute) in order to avoid transient and hysteretic effects (although a small hysteresis is always present in OECTs). We always measured from the on-state to the off-state. The procedure has been added to the Methods. The leakage is only visible in log scale, and becomes comparable to the drain current only in the off state, below threshold, where we do not claim that our model is valid, as discussed above.

13. Methods. More details about the device fabrication must be provided. Were the electrodes patterned through lift-off or RIE etching? How was the channel patterned (RIE)? Was there a post-annealing step? What SMU was used for the measurements? How was the volumetric capacitance measured? Figure 4a-e shows data for OECTs using liquid electrolytes. But in methods, measurements using solid electrolytes is also mentioned.

Following your suggestions, we strongly improved the Methods section, including the answers to your questions and many more details about fabrication and measurements.

14. Although the entropic contributions seem to well describe steady state behavior of the presented OECT, the manuscript fails to communicate its transient behavior. What are the limitations of the current approach?

We purposefully avoided discussing transient behavior. It is really difficult to decouple thermodynamic processes (where only initial and final states matter) and kinetic mechanisms.

Going beyond steady state is the objective of current work and future publications. We do not claim that the thermodynamic approach automatically translates to transient dynamics. It needs to be extended and kinetic processes must be included. We now included this in section 3:

Moreover, decoupling thermodynamic and kinetic effects is a challenging task. To this end, each voltage sweep was performed very slowly (1 V/min) to ensure that the system is always at equilibrium. This is a "static" models, and the description of transient effects may go beyond the thermodynamic approach.

How does the method compare against the approximations put forward by ref 25 (fig 2b,c,f)?

In Ref 25, Tybrandt et al use a different approach, based on electrostatics. There, the parameters that is voltage-dependent is the mobility, which we assumed constant for simplicity. As such it is really difficult to compare the two models and one may imagine that both mobility and capacitance are voltage-dependent. We have a model that can describe experimental findings very well without assuming specific transport properties of the material. Of course, mobility might depend on the ion concentration, ion size etc. which could be included in the model to get more accurate fitting.

15. The entropic term is mainly determined by ϕ . The manuscript doesn't clearly suggest a trajectory for improving the device performance. In light of the presented results, how can the materials be tailored for different applications? For example, high gain vs high speed?

We now added a final section 5 "Implication for device and material design" outlining how the thermodynamic model offers pathways for improvement. Please read above

How to use this framework to compare among materials based on the proposed model?
Also now outlined in the section 5

Finally, we believe a more succinct title such as "thermodynamics of organic electrochemical transistors" is more clear and appropriate

REVIEWERS' COMMENTS

Reviewer #1 (Remarks to the Author):

I find the manuscript much improved in this revision, and many questions and ambiguities have been answered. Considering the degree of novelty of this interpretation, and the potential impact of the thermodynamic interpretation, I support publication in the present form.

Reviewer #2 (Remarks to the Author):

Authors have done an excellent job in addressing all the comments raised in the previous review. There are a couple of minor corrections before publication.

1. SI figure 3. Unit of current is in A, and scale is linear. This is incorrect. Please revise.
2. Please include a micrograph of the device in SI.
3. Authors have satisfactorily justified the use of linear scale for transfer curves. Nevertheless, it is still advised to show at least one figure in SI in log scale to show the deviation in the subthreshold region for future researchers.

REVIEWERS' COMMENTS

Reviewer #1 (Remarks to the Author):

I find the manuscript much improved in this revision, and many questions and ambiguities have been answered. Considering the degree of novelty of this interpretation, and the potential impact of the thermodynamic interpretation, I support publication in the present form.

Our answer: We thank you for the precious feedback provided

Reviewer #2 (Remarks to the Author):

Authors have done an excellent job in addressing all the comments raised in the previous review.

We thank you a lot for the precious feedback. Please find below our response to the latest remarks

There are a couple of minor corrections before publication.

1. SI figure 3. Unit of current is in A, and scale is linear. This is incorrect. Please revise.
The figure has been corrected. The unit should be indeed mA, but the linear scale is correct.

2. Please include a micrograph of the device in SI.

Now added

3. Authors have satisfactorily justified the use of linear scale for transfer curves. Nevertheless, it is still advised to show at least one figure in SI in log scale to show the deviation in the subthreshold region for future researchers.

We agree. We added now in the SI the log scale version of those curves.